# Acute Coronary Syndrome Treated with Percutaneous Coronary Intervention in Hutchinson–Gilford Progeria

**DOI:** 10.3390/children10030526

**Published:** 2023-03-08

**Authors:** Luciano De Simone, Serena Chiellino, Gaia Spaziani, Giulio Porcedda, Giovan Battista Calabri, Sergio Berti, Silvia Favilli, Laura Stefani, Giuseppe Santoro

**Affiliations:** 1Paediatric Cardiology, “Meyer” Children’s Hospital, University of Florence, Viale Pieraccini, 24, 50139 Florence, Italy; 2Interventional Cardiology, “Heart Hospital”, National Research Council—Tuscany Foundation “G. Monasterio”, 54100 Massa, Italy; 3Sports Medicine Center, Clinical and Experimental Department, University of Florence, 50134 Florence, Italy; 4Pediatric Cardiology and GUCH Unit, “Heart Hospital”, National Research Council-Tuscany Foundation “G. Monasterio”, 54100 Massa, Italy

**Keywords:** Hutchinson–Gilford progeria, acute coronary syndrome, interventional cardiac catheterization, stent implantation, revascularization

## Abstract

Hutchinson–Gilford progeria syndrome is an extremely rare genetic disease caused by a de novo mutation in the LMNA gene, leading to an accumulation of a form of Lamin A, called Progerin, which results in a typical phenotype and a marked decrease in life expectancy, due to early atherosclerosis and cardiovascular disease. We report the case of a fourteen-year-old Chinese boy with Hutchinson–Gilford progeria syndrome admitted to the emergency room because of precordial pain. Physical examination showed tachycardia 130 beats/min and arterial hypertension: 170/120 mmHg, normal respiratory rate, no neurological impairment; ECG evidenced sinus tachycardia, left ventricular hypertrophy, horizontal ST-segment depression in I, aVL, II, III, aVF leads, and V4–V6 and ST-segment elevation in aVR and V1 leads. Echocardiography highlighted preserved global left ventricular function with concentric hypertrophy, altered diastolic flow pattern, mitral valve insufficiency, and minimal aortic regurgitation. Blood tests evidenced an increase in high-sensitivity troponin T level (335 pg/mL). NSTEMI diagnosis was performed, and the patient was admitted to the intensive care unit. A coronary CT angiography showed a severe obstruction of the common trunk of the left coronary artery, for which an urgent percutaneous coronary intervention (PCI) was proposed. A selective coronary angiography imaged complete chronic occlusion of the left main coronary artery as well as severe stenosis at the origin of a very enlarged right coronary artery that vascularized the left coronary artery through collaterals. Afterwards, the right coronary artery was probed using an Amplatz right (AR1) guiding catheter, through which a large 3.5 mm drug-eluting coronary stent (Xience Sierra, Abbott, Abbott Park, IL, USA) was implanted. At the end of the procedure, no residual stenosis was imaged and improved vascularization of the left coronary artery distribution segments was observed. Dual antiplatelet therapy (DAPT) consisting of aspirin (75 mg daily) and clopidogrel (37.5 mg daily) and anti-hypertensive therapy were started. At the one-year follow-up, the patient had not reported any occurrence of anginal chest pain.

## 1. Introduction

Hutchinson–Gilford progeria syndrome is an extremely rare human genetic disease caused by a de novo heterozygous single base substitution mutation at codon 608 (G608G: GGC > GGT) in LMNA, in approximately 90% of patients [1,2]. This gene codes for Lamin-A and Lamin-C proteins. The defective prelamin A processing gives rise to the accumulation of a dominant negative form of Lamin-A, called progerin [3,4]. Patients with Hutchinson–Gilford progeria syndrome show severe failure to thrive after one year of life. Phenotypic hallmarks of Hutchinson–Gilford progeria syndrome include low weight and short stature, premature aging with alopecia and sclerotic skin and craniofacial disproportion, sleeping with open eyes, bone alterations associated with joint contractures and prominent cutaneous vasculature, laboratory findings such as prolonged prothrombin times, high platelet counts, and elevated serum levels of phosphate [5,6,7], but most of all the mutant progerin causes extensive atherosclerosis and consequent cardiovascular and neurological involvement leading to premature death within the second decade of life [8]. The most common cardiac findings at autopsy include coronary and cerebrovascular atherosclerosis, left ventricular hypertrophy, mitral-aortic thickening and calcification, myocardial necrosis, and fibrosis [9,10,11]. Clinical manifestations include arterial hypertension, angina pectoris, myocardial infarction, and stroke [12]. Electrocardiography (ECG) is a sensitive tool in Hutchinson–Gilford progeria syndrome, showing ventricular repolarization abnormalities such as ST segment depression and T wave inversion or flattening [13,14], and echocardiography is universally used to assess left ventricle global and segmental kinetics. Recently, coronary CT angiography [15,16] has proven effective in non-invasively diagnosing coronary lesions, with high sensitivity and diagnostic specificity in adults. We describe the case of a 14-year-old Chinese boy suffering from Hutchinson–Gilford progeria syndrome admitted to the emergency department with precordial pain suggestive of ischemia and hypertensive crisis.

## 2. Case Description

A 14-year-old boy followed at our hospital because of Hutchinson–Gilford progeria syndrome was admitted to the emergency room because of anginal precordial pain. In his past medical history, we underlined arterial hypertension, ischemic stroke, abdominal surgery for malrotation, and, more recently, severe osteoporosis and Dupuytren’s syndrome; over the last four months chest pain was reported during emotional changes without fever or dyspnea. On physical examination, the patient was apiretic and appeared to be suffering from chest pain and tachycardia, with a heart rate of 130/min and hypertensive blood pressure: 170/120 mmHg. In order to exclude evaluation errors, the blood pressure was measured in both arms after a few minutes with the patient sitting in a quiet environment, using a standard auscultation technique and size appropriate cuffs. We also observed normal heart sounds, regular breathing rate, and normal chest auscultation.

The abdomen appeared rounded and tense, but no alterations in abdominal palpation and percussion were detected. He showed the characteristic findings of progeria, such as short stature: 126 cm (<3rd centile) and low weight: 14 kg (<3rd centile), with body mass index (BMI) 8.8 kg/m^2^. Given the severe growth restriction typical of this syndrome, we considered chronological age instead of the height–age ratio for the calculation of blood pressure percentiles related to sex. Additionally, he had sclerotic skin without subcutaneous fat and joint stiffness, craniofacial disproportion, thin lips, micrognathia, clavicular hypoplasia, and alopecia, but no neurological or cognitive impairment. Blood tests evidenced an increase in high-sensitivity troponin T (335 pg/mL), dyslipidemia total cholesterol 105 mmol/L, low-density lipoprotein (LDL) 53 mmol/L, triglycerides 237 mmol/L, and HDL 25 mmol/L, with normal thyroid function and glycaemia. Instrumental findings: the 12-lead. ECG, which was applied according to a standard method and recorded during precordial pain, showed sinus tachycardia, normal axis, left ventricular hypertrophy, horizontal ST-segment depression in I, aVL, II, III, aVF leads, and V4–V6 and ST-segment elevation in aVR and V1 leads, which later normalized (Figure 1A,B). Electrocardiographic intervals were corrected for heart rate using the Bazett formula. Transthoracic echocardiography was performed by applying standard imaging techniques, according to the American Society of Echocardiography, and showed preserved global left ventricular function with concentric hypertrophy, except for a small area of thinning and dyskinesia of the apical interventricular septum, altered diastolic flow pattern, severe mitral and mild aortic regurgitation, and no pericardial effusion with apparent normal coronary artery implantation and initial course. NSTEMI diagnosis was performed, and the patient was admitted to the ICU at the Meyer Children’s Hospital in Florence. 

The patient showed a good response to intensive therapy with intravenous nitrates and calcium channel blockers, with disappearance of pain, blood pressure normalization, and improvement of the ST segment modifications at ECG (Figure 1B). After the acute phase, a coronary CT angiography was performed. A severe obstruction of the common trunk of the left coronary artery was observed due to plaque formation, with phenotypic characteristics similar to arteriosclerosis occurring in geriatric patients, including calcification and evidence of plaque erosion. Ascending aorta calcification was also observed, with regular diameters. Low-dose oral antiplatelet, high-potency statin, and anti-hypertensive drugs were started. A multidisciplinary team, including pediatric cardiologists, pediatricians, an adult cardiologist, and a coronary interventionalist, decided to have the patient undergo PCI in a pediatric cardiology and cardiac surgery center (IFC-CNR Massa Cardio-thoracic Centre). Interventional cardiac catheterization was performed from the right femoral artery. Aortic angiography imaged complete chronic occlusion of the left main coronary artery that was severely calcified at the origin, as well as ninety-five percent stenosis at the origin of the right coronary artery, which perfused the anterior interventricular coronary artery through collateral circulation (Figure 2). This vessel did not show any further significant stenosis downstream and completely vascularized the left ventricular segments supplied by the left coronary artery. The right coronary artery was probed using an Amplatz right (AR1) guiding catheter, through which a large 3.5 mm drug-eluting coronary stent (Xience Sierra, Abbott, USA) was implanted (Figure 3). At the end of the procedure, no residual stenosis and improved revascularization of the left coronary artery distribution segments was imaged (Figure 4). No attempt was made to probe the left main coronary artery due to severe calcification of the origin as well as a complete supply of its distribution territories by a very enlarged right coronary artery. A few days after intervention, the patient was transferred again to Meyer Children’s Hospital in Florence and discharge was possible after one week. Dual antiplatelet therapy (DAPT) was prescribed, with a combination of aspirin (75 mg daily) and clopidogrel (37.5 mg daily) and anti-hypertensive therapy.

After the intervention, a dramatic improvement of both cardiac and systemic symptoms was observed; at echocardiographic controls, the abnormalities of the left ventricular motion had disappeared, while left ventricular diastolic dysfunction [17] was still detected. The patient presented a prompt response with a good mid-term outcome. At the one-year follow-up, the patient reported no occurrence of anginal chest pain.

## 3. Discussion

Most studies on Hutchinson–Gilford progeria syndrome are epidemiological or oriented towards the description of the phenotypic characteristics; some others focus on the autopsy findings or the changes in laboratory tests and more recently on the genetic aspects. So far, relatively few studies have dealt with cardiac complications, and these mainly describe the results of electrocardiography and echocardiography [18,19,20,21]. The causes of death in Hutchinson–Gilford progeria syndrome are mainly due to cardiovascular events, and death usually occurs in the second decade of life from myocardial infarction or cerebrovascular disease caused by premature arteriosclerosis; the average life expectancy for a child with progeria is 13–14 years [22]. However, for many years the therapeutic aspects were little considered; a premature death was reported as an inevitable event and no treatment advices have so far been proposed, either for the chronic management or for the acute complications. Only in recent years have a few anecdotal cases been reported, in which an emergency therapy or cardiac surgery was performed [23,24]. The most interesting aspect of this case consists of a PCI attempted in near emergency conditions with a high risk coronary circulation, which proved to be successful. This patient was already known due to a previous stroke and dyslipidemia; the latter, not always described in Hutchinson–Gilford progeria syndrome, may have been an additional risk factor for cardiovascular events [25]. In the emergency room we observed a short duration of symptoms and a slight increase in the troponin level. The ECG pattern, however, showed extensive ischemic involvement of the left ventricular anterior wall, although reversible, as already described in a Hutchinson–Gilford progeria syndrome patient [26] in the context of an echocardiographic pattern of diastolic left ventricular dysfunction, preserved contractility, and concentric hypertrophy. The image of the septum thinning was probably due to a previous ischemic event, but the small area of dyskinesia of the apical septum must be interpreted as a sign of acute ischemic distress. Therefore, the decision to first submit the patient to a CT angiography, as a safer procedure, can be shared [27,28]; in this way we could evidence the lesion of the common trunk of the left coronary artery, which appeared to be very critical. The decision to continue in the diagnostic investigation with angiography was based on the observation of extensive electrocardiographic involvement which could not be explained merely by the ostial lesion of the left coronary artery, which appeared to be chronic. On the basis of the ECG, involvement suspicion could be raised of a lesion of the right coronary one, potentially eligible for PCI. Indeed, the coronary angiography evidenced, in addition to the left coronary lesion, a severe proximal stenosis of the right coronary artery, not detected by angio-CT. We hypothesize that this might be due to motion artifacts related to high heart rate, a known limitation of coronary angio-CT in children [29]. The critical involvement at the origin of the two main coronary arteries is a life-threatening condition and usually requires surgical treatment in adult patients [30]. However, this was deemed too risky in this precarious patient because coronary artery bypass grafting (CABG) has rarely been described in pediatric patients, and then mainly in the treatment of coronary complications of Kawasaki disease, with only one case in a progeria patient, who died during surgery [24]. Thus, based on risk-benefit assessment, treatment with PCI was deemed the best therapeutic option. To the best of our knowledge, only one previous case of successful PCI has been described in a 13-year-old boy with progeria in an acute myocardial infarction [23]. Hence, this represents the first case of successful PCI procedure in a Hutchinson–Gilford progeria syndrome patient reported in the literature as an elective procedure. The particular condition of the disease creates atherosclerotic alterations similar to those of the adults in a small arterial tree, which makes its vascular access more complex, requires the use of miniaturized catheters and balloons, and increases the risk of complications such as arterial dissection [31]. This case confirms the possibility of applying percutaneous treatment of coronary atherosclerotic lesions even in low weight pediatric patients and with a complex coronary tree. Furthermore, it proves that electrocardiography still plays a central role in the diagnosis of myocardial ischemia as the ST changes have proven to be the earliest test in guiding the therapeutic options. The use of dual antiplatelet therapy, recommended in adult patients submitted to PCI for secondary prevention of ischemic events but still not reported in children, has proven to be effective and safe in this patient in a mid-term follow-up [32]. 

## 4. Conclusions

Hutchinson–Gilford progeria syndrome is a representative disease of premature aging, and patients suffer from decreased life expectancy and impaired quality of life. The premature aging processes, especially accelerated arteriosclerotic, play a key role in the onset of cardiovascular events, which are the main cause of death in these patients. It is still unclear whether the mechanisms involved in these children are the same as those seen in typical adult arteriosclerosis; further research is needed to clarify these aspects

Cardiac assessment is recommended in all patients with Hutchinson–Gilford progeria syndrome, as well as periodic cardiac follow-up. Treatment of coronary artery disease in Hutchinson–Gilford progeria syndrome patients remains challenging due to lack of guidelines and clinical trials. This case report proves that recovery from acute coronary syndrome through percutaneous angioplasty is also possible in children and adolescents with Hutchinson–Gilford progeria syndrome. Clinical management and therapeutic strategies must be improved in this pediatric population, so further studies are required.

## Figures and Tables

**Figure 1 children-10-00526-f001:**
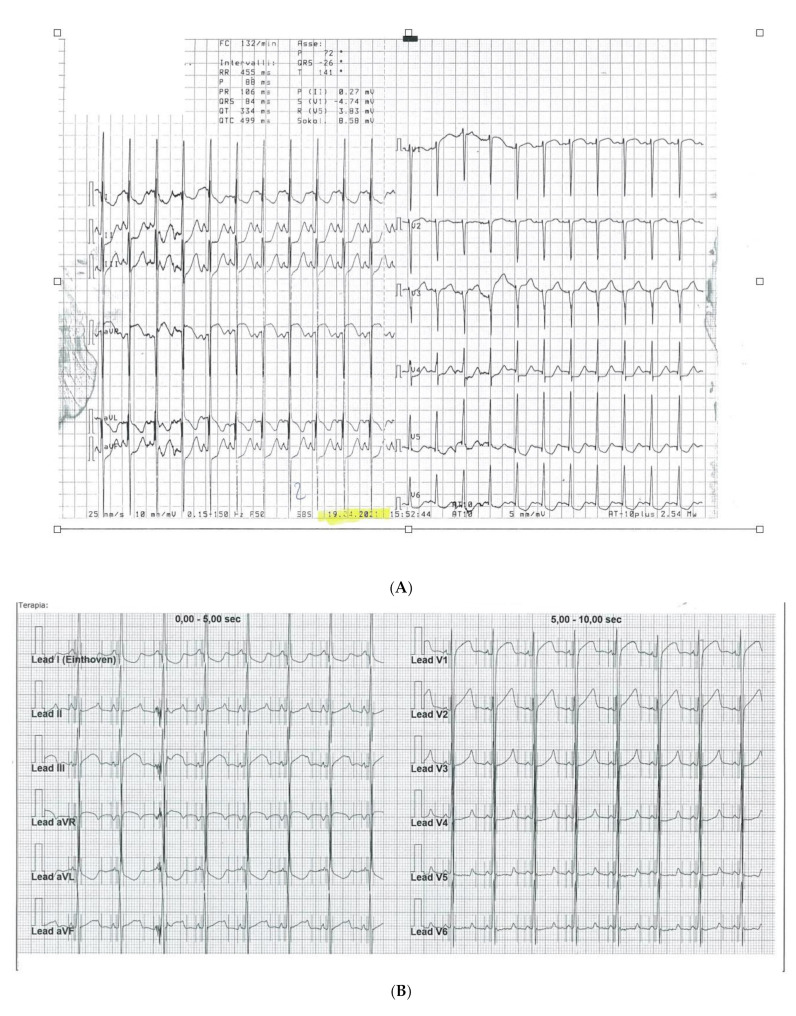
(**A**). ST depression in infero-lateral leads during chest pain. (**B**). Improvement of ST depression and increased R-wave voltage V1–V3 at disappearance of symptoms.

**Figure 2 children-10-00526-f002:**
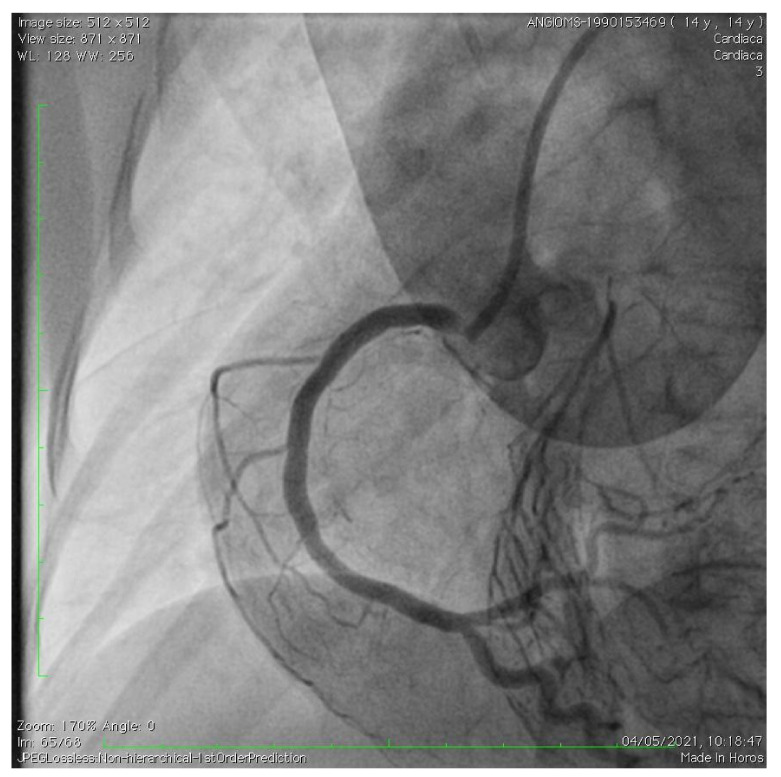
RCA angiography: severe ostial stenosis and imaging of collateral circles with left coronary artery.

**Figure 3 children-10-00526-f003:**
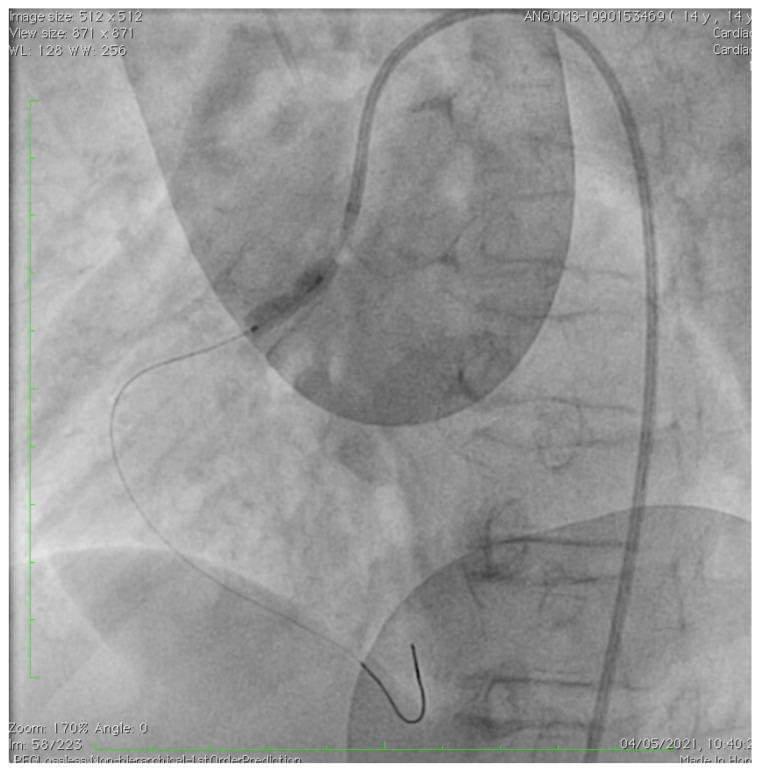
Stent implantation at the ostium of RCA.

**Figure 4 children-10-00526-f004:**
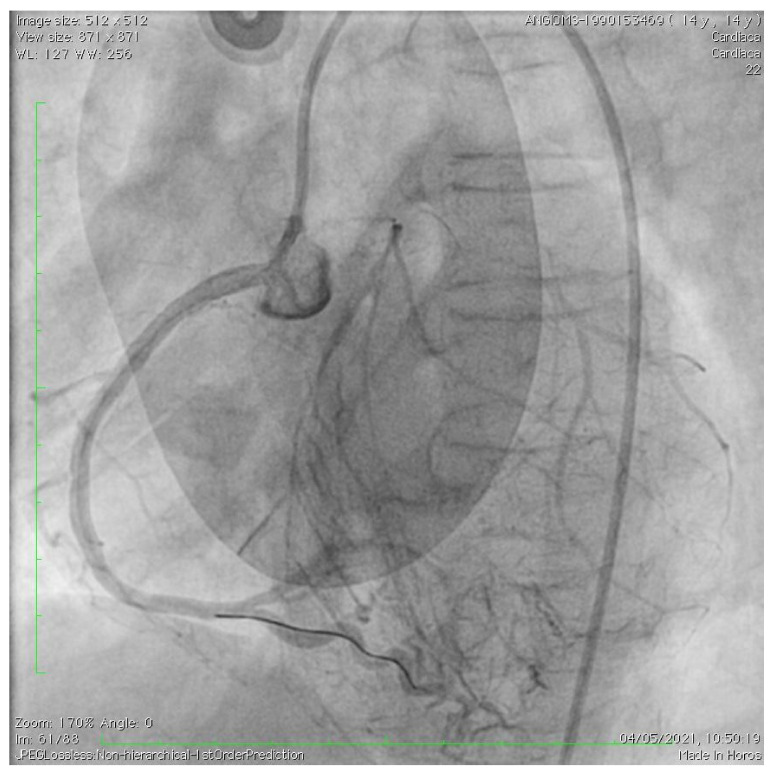
After PCI, no residual stenosis of proximal RCA.

## Data Availability

the data presented in this study are openly available in Cardiology Unit of A. Meyer Hospital Florence and Cardiology Unit of Heart Hospital National Research Council-Tuscany Foundation “G. Monasterio”, Massa, Italy.

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
