# Peer review of "Acute Coronary Syndrome Treated with Percutaneous Coronary Intervention in Hutchinson–Gilford Progeria"

_children, 2023, doi:10.3390/children10030526_

Round 1

Reviewer 1 Report

I would like to thank the authors for presenting this interesting case.
I have some comments and recommendations.

1- The term PTCA used in the title and throughout the article usually refers to balloon angioplasty of the coronaries and not stent implantation; the latter is referred to by the term "PCI" which stands for percutaneous coronary intervention. So, I suggest using the term "PCI" because it is more scientifically sound, while the term "PTCA" will create confusion, especially among coronary interventionists who rarely use balloon angioplasty alone without stents.
2- The syndrome has been given the abbreviation HGPS in the first line of the introduction. Yet, the authors used the same long name and not the abbreviation throughout the article.
3- Some expressions are used in the articles which I could not understand or find a meaning for: 
"Authoptic" (line 56) has no meaning in the dictionary

"TC" (lines: 33, 92, 146): I think the authors mean CT which stands for computed tomography; they used the correct expression in line 136
3- The authors used 2 synonyms in the article (insufficiency and regurgitation) (line 84). It is recommended to stick to one term.
4- The authors describe the coronary lesion as having evidence of plaque erosion or rupture (line 96). It is known that those are the 2 mechanisms for plaque instability and acute coronary syndrome, so only one should be present; i.e. they do not co-exist. Besides, the authors did not provide us with the methods they used to diagnose those underlying mechanisms, which usually entail using intracoronary imaging (e.g. IVUS or OCT).
5- In line 120, the authors state that the left ventricular diastolic dysfunction was the most important echo abnormality after the intervention. Does this mean that the wall motion abnormalities that were present disappeared? Usually diastolic dysfunction is not regarded as an important finding unless it is of a severe degree, was that the situation with this patient?
6- In lines 136-137: the authors state that they had to do CT angiography because coronary angiography was not recommended in elective cases. But actually, this was not an elective case. it was a non-STE acute coronary syndrome, in which coronary angiography was highly indicated according to the guidelines and not the CT.
7- In line 145: the authors stated that the lesion in the ostium of the right coronary artery was discovered in coronary angiography but not detected by the CT. This is very strange because CT is a very good tool for diagnosing ostial lesions; even better than coronary angiography because of the catheter-induced spasm that may happen at the ostium. So, the authors should explain why this happened. Was it because of motion artifacts due to high heart rate for instance?
8- Line 72: The word "afebrile" is repeated twice. Line 74: the word "objectivity" is out of context; it does not fit in the sentence.

Author Response

Dear Reviewer , I have uploaded  the revised manuscript .You'll find  all your requested changes in red. In addition we implemented discussion and conclusions Best regards.
Dr. Luciano De Simone

Reviewer 2 Report

Dear authors,

The case is really interesting. The authors explain well the case with the results and de discussion. I think the case is really important to progeria field to develop or research in new therapies to treat this disease.